# OUT-OF-DISTRIBUTION DETECTION WITH DIFFUSION-BASED NEIGHBORHOOD

## ABSTRACT

Out-of-distribution (OOD) detection is an important task to ensure the reliability and safety of deep learning and the discriminator models outperform others for now. However, the feature extraction of the discriminator models must compress the data and lose certain information, leaving room for bad cases and malicious attacks. In this paper, we provide a new assumption that the discriminator models are more sensitive to some subareas of the input space and such unfair treatment creates bad cases and overconfidence areas. Under this assumption, we design new detection methods and indicator scores. For detection methods, we introduce diffusion models (DMs) into OOD detection. We find that the denoising process (DDP) of DMs also functions as a novel form of asymmetric interpolation, which is suitable to enhance the input and reduce the overconfidence areas. For indicator scores, we find that the features of the discriminator models of OOD inputs occur sharp changes under DDP and use the norm of this dynamic change as our indicator scores. Therefore, we develop a new framework to combine the discriminator and generation models to do OOD detection under our new assumption. The discriminator models provide proper detection spaces and the generation models reduce the overconfidence problem. According to our experiments on CIFAR10 and CIFAR100, our new methods successfully outperform state-of-the-art methods. Our implementation is put in the supplementary materials.

## 1 INTRODUCTION

Out-of-distribution (OOD) detection is an important task for deep models that helps the models determine their capability boundary and keep them from being fooled by OOD data. It has a strong connection with many real-world machine-learning applications, such as cybersecurity (Xin et al., 2018), medical diagnosis (Latif et al., 2018; Guo et al., 2020) and autopilot (Geiger et al., 2012). The existing methods for OOD detection can be generally categorized into discriminator-based and generation-based methods. The discriminator-based methods (Wang et al., 2022) use the logit or the feature space to do that. The generation-based methods (An & Cho, 2015; Nalisnick et al., 2019) use the reconstruction difference in data space or density estimation in latent space to do that.

The discriminator-based methods can extract useful features and make the detection faster and better in most cases. However, such extraction and compression lose some information and leave room for bad cases and malicious attacks (Goodfellow et al., 2014; Amodei et al., 2016). The generation-based methods can capture the whole data distribution but lack effective indicator scores to compete with the SOTA discriminator-based methods, partly because of the curse of dimensionality. Previous works mostly concentrate on solving these challenges using only one kind of model. For discriminator-based methods, Wang et al. (2022) combine the information from both features and logits. Sehwag et al. (2020) use self-supervised learning to improve feature extraction. For generation-based methods, Nalisnick et al. (2019) use the typicality set to design better indicator scores. Jiang et al. (2022) use statistical methods in the latent space, such as the Kolmogorov-Smirnov test.

In addition to overcoming the problems of each kind of model by itself, we find that generative and discriminative models can be combined and solve each other's problems. We provide a new assumption that the discriminator models are more sensitive to some subareas of the input space to explain the existence of bad cases and overconfidence areas. To solve this problem, we get inspiration from water quality detection in the real world. They use some fixed detectors and make the water

*explain our motivation*

flow by stirring. Then these detectors can monitor a large area of water. For OOD detection, the discriminator models also concentrate on some fixed subareas of the whole input space. Therefore, we also want to "stir" the input to improve detection accuracy and reduce overconfidence areas. We find that generation models are a good choice to be the "stirring" operators. Under such operators, the results of InD data remain normal at all times and that of OOD data expose anomalies.

To design suitable generation strategies that can enhance discriminator models, we introduce diffusion models (DMs), which play an important role in generation models, into OOD detection. DMs have created many state-of-the-art generation results, including (Vahdat et al., 2021; Ho et al., 2022). We dive into the structure of DMs and find that the diffusion denoising process (DDP) of DMs can be an ideal choice for the "stirring" operator we mentioned above. Because it can adjust any level of feature space and provides tools to keep the adjustment under control by using the denoising and interpolation properties. Such an operator "stirs" the input and needs to be resampled several times to make the result convergent, which builds a neighborhood of input data, called the diffusion-based neighborhood (DiffNB). According to our above analysis, the feature of OOD data explores anomalies, which means that the feature can change sharply. We can detect such anomalies by simple Euclidean distance between several different features of DiffNB. Our pipeline is in Figure 1.

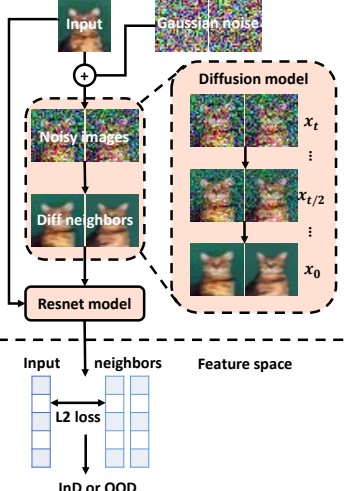

Figure 1: The pipeline of our detection method.

We choose ten representative methods to compare with our methods on two representative datasets: CIFAR10 and CIFAR100. According to our experiments, our new methods outperform existing models and methods in most cases. Our work has the following contributions:

- We provide a new assumption that the discriminator models are more sensitive to some subareas for OOD detection. We analyze why it causes the overconfidence problem and how to solve it.

- We find that the diffusion denoising process of invertible diffusion models is a novel kind of asymmetric interpolation, which can keep the InD data relatively unchanged and provide tools to control the direction of the denoising process.

- We develop a framework to combine the discriminator and generation models, which uses a ResNet to extract features and the diffusion denoising process of a diffusion model to reduce overconfidence areas. Our methods get competitive OOD detection results with SOTA methods.

## 2  BACKGROUND

In this section, we first introduce existing methods for OOD detection. Then, we show the development of diffusion models related to our paper. Because of the limited space in the main paper, more related works about diffusion models can be found in Appendix A.1.

### 2.1  OUT-OF-DISTRIBUTION DETECTION

OOD detection is an important task that can help neural networks to determine their capability boundary. More specifically, let $X = \{x_1, \ldots, x_n\} \sim p$ be a group of images from the in-distribution (InD) $p$. We want to build a detector $f$ that $f(x_1, \ldots, x_n) = 1, \forall i, p(x_i) \geq \sigma$ and $f(x_1, \ldots, x_n) = 0, \forall i, p(x_i) \leq \sigma$. Here, $\sigma$ controls the decision boundary. When we get another group of data $Y = \{y_1, \ldots, y_n\}$, we decide whether this group is from InD $p$ or an unknown distribution $q$ based on the results of $f$. If $n = 1$, this is pointwise OOD detection, and if $n \geq 2$, this is group OOD detection. In general, the existing OOD detection methods can be categorized into discriminator-based and generation-based methods.

Discriminator-based methods design indicator scores based on the output of discriminator models. Some methods can be used without modifying the model. ODIN (Liang et al., 2018) uses temperature scaling and the softmax results to detect OOD samples. ViM (Wang et al., 2022) combines the information of features and logits. KNN (Sun et al., 2022) includes the kth nearest neighbor of the

input in feature space into the detection process. Some methods try to improve the detection ability in the training process. G-ODIN (Hsu et al., 2020) designs a new loss function. ConfGAN (Sricharan & Srivastava, 2018) generates OOD data using GANs to help the discriminator models to determine the boundary. PixMix (Hendrycks et al., 2022) uses data augmentation to improve the detection results. SSD (Sehwag et al., 2020) uses self-supervised learning to improve feature extraction.

Generation-based methods use the reconstruction difference in the input space and the density estimation in the latent space to do OOD detection. An & Cho (2015) use the reconstruction ability of VAEs. Some methods assume that the generation models can reconstruct the in-distribution data better. Some methods use the Distribution transformation capability of generation models and transfer the input distribution into simple Gaussian distribution. The likelihood of the input becomes a direct choice, but Nalisnick et al. (2018) finds that OOD data can also locate in the high-likelihood area. Nalisnick et al. (2019) find the InD data is concentrated in the typical set instead of the high likelihood area and design new methods using the typical set. Serrà et al. (2019) find that we can use input complexity to correct the bias of likelihood. In addition to the likelihood, many existing statistical methods can detect whether a distribution obeys standard Gaussian distribution. Zhang et al. (2020) uses KL-divergence to detect OOD data. Jiang et al. (2022) use a nonparametric statistics method called the Kolmogorov-Smirnov test.

## 2.2 DIFFUSION MODEL

**Classical diffusion model**  DMs build a transformation from Gaussian distribution to image distribution through a multistep denoising process. Given a data distribution $x_0 \sim q(x_0)$, the diffusion process satisfies a Markov process as following Ho et al. (2020):

$$q(x_{1:T}|x_0) = \prod_{t=1}^{T} \mathcal{N}(\sqrt{1-\beta_t}x_{t-1}, \beta_t I)$$
$$q(x_t|x_0) = \mathcal{N}(\sqrt{\bar{\alpha}_t}x_0, (1-\bar{\alpha}_t)I). \tag{1}$$

Here, $T = 1000$, which is the max iteration step. $\beta_t \in (0,1)$, which controls the speed of adding noise. Additionally, $\alpha_t = 1 - \beta_t$, $\bar{\alpha}_t = \prod_{i=1}^{t} \alpha_i$, $\bar{\mu}_t = \frac{\sqrt{\bar{\alpha}_{t-1}}\beta_t}{1-\bar{\alpha}_t}x_0 + \frac{\sqrt{\alpha_t}(1-\bar{\alpha}_{t-1})}{1-\bar{\alpha}_t}x_t$ and $\bar{\beta}_t = \frac{1-\bar{\alpha}_{t-1}}{1-\bar{\alpha}_t}\beta_t$. The objective function is defined by:

$$L_{t-1} = \mathbb{E}_{x_0,\epsilon}\left[\frac{\beta_t^2}{\alpha_t(1-\bar{\alpha}_t)}||\epsilon - \epsilon_\theta(\sqrt{\bar{\alpha}_t}x_0 + \sqrt{1-\bar{\alpha}_t}\epsilon, t)||^2\right]. \tag{2}$$

Here, $\epsilon_\theta$ is an estimate of the noise $\epsilon$. After we get well-trained $\epsilon_\theta$, according to Song et al. (2021a), the denoising process of Denoising Diffusion Probabilistic Models (DDPMs) and Denoising Diffusion Implicit Models (DDIMs) satisfies:

$$x_{t-\delta} = \sqrt{\bar{\alpha}_{t-\delta}}\left(\frac{x_t - \sqrt{1-\bar{\alpha}_t}\epsilon_\theta(x_t, t)}{\sqrt{\bar{\alpha}_t}}\right) + \sqrt{1-\bar{\alpha}_{t-\delta} - \sigma_t^2}\epsilon_\theta(x_t, t) + \sigma_t\epsilon_t. \tag{3}$$

Here, $\delta$ is the iteration step size. If $\sigma_t$ equals one, Equation (3) represents the denoising process of DDPMs; if $\sigma_t$ equals zero, this equation represents the denoising process of DDIMs. In Appendix A.3, we will further describe how to make the iteration of the diffusion model fast and invertible.

**Classifier-free guidance**  Ho & Salimans (2021) show a simple and effective way to generate conditional samples called classifier-free guidance. It adds a condition embedding $c$ into $\epsilon_\theta$ in the training process and changes the final estimation of noise as:

$$\bar{\epsilon}_\theta(x_t, c) = (1+\omega)\epsilon_\theta(x_t, t) - \omega\epsilon_\theta(x_t). \tag{4}$$

Here, $\omega$ is the guidance weight, which controls the balance between realness and diversity.

## 3 DIFFUSION-BASED NEIGHBORHOOD FOR OOD DETECTION

In this section, we first theoretically define our task, which is to design proper additional operators for a fixed detector. We assume that the detector only concentrates on some subarea of the input

space. Then we use a toy example to show how to design proper additional operators and find that reconstruction operators are potential candidates for general cases. After that, we use invertible diffusion models and the diffusion denoising process (DDP) to develop a new asymmetry interpolation to satisfy the requirements of the additional operators. Then we use a new toy example to show what will happen under DDP for more complex situations and use the norm of the dynamic change under DDP as our detection score. Both DDP and our new score can be applied to any black box detector $f$ and we introduce classifier-free guidance for the multi-class condition. In the end, we provide our general methods and visualize their actual effect when the detector is a pre-trained ResNet18.

## 3.1 THEORETICAL SETTING

We first transfer the OOD detection task and the idea behind water quality detection mentioned in Section 1 into a theoretical problem. Following the definitions in Section 2.1, the detector is a function $f$ that can be an analytic function or a neural network. The detection problem is that given a distribution $p$ on a certain space $U$, let $\{x \in U | p(x) > \sigma\}$ be $W$ and we want to find a detector $f$ that satisfies $f^{-1}(1) = W$ exactly. The idea behind water quality detection is to use a fixed detector and make water flow by stirring. The fixed detector means that the detector $f$ may treat bad cases as normal and $f^{-1}(1)$ is strictly bigger than $W$. The stirring operator means that we can add additional operators to help the detection. Therefore, we can define a group of operators $\{g_i\}_{i \in I_g}$[1] satisfying $f \circ g_i(x) = f(x) = 1$ if $x \in W$ and $\cap_{i \in I_g}(f \circ g_i)^{-1}(1) \subset f^{-1}(1)$. The first condition means that all $f \circ g_i$ can identify the InD data correctly and the second condition means that the $\cap_{i \in I_g}(f \circ g_i)^{-1}(1)$ becomes smaller by adding more $g_i$. Now we introduce the theoretical version of our problem:

> *Given a fixed subarea $W \subset U$, a single value function $f$ satisfying $W \subset f^{-1}(1)$,*
> *how can we design operators $\{g_i\}_{i \in I_g}$ to minimize $\cap_{i \in I_g}(f \circ g_i)^{-1}(1)$?*

For our task, $f$ is a neural network and the difference between $f^{-1}(1)$ and $W$ is the overconfidence area. Here, we assume all training data belong to one class. To design proper $\{g_i\}_{i \in I_g}$, we need to locate what causes such an overconfidence problem. We find that discriminator models are similar to the fixed water detector and only concentrate on some subareas of the whole input space. Here, we provide the main assumption in our paper:

> *For a discriminator model, different channels concentrate on different subareas*
> *of the input space. Some subareas are more important than others.*

Here, we emphasize that even in the class-related area, some subareas are more important than others. To show why this assumption is reasonable, We dive into the structure of a discriminator model and find that the linear operators, including the convolutional layers and the following pooling layers, play an important role. According to the singular value decomposition, every linear operator $L_A : \mathbb{R}^n \to \mathbb{R}^m$, $m < n$, can be transferred to a restriction operator[2] $L_{A*}|_S$ under proper change of basis and $S$ is a $m$-dimensional subspace. The input data is bounded in our task, so the subspace $S$ becomes a bounded area in the input space. Different linear kernels in a layer concentrate on different $S_i$ and the total sensitive area of this layer is $\cup_{i \in I_S} S_i$. Some subareas can be included by more $S_i$, which are more important than the remaining area.

## 3.2 SINGLE-LAYER MODEL

The whole input space may be covered by the support area of all kernels $\cup_{i \in I_S} S_i$ and some areas are covered more times than others. For simplification, we can transfer the problem to a restriction operator. The key idea is to set a threshold $N$ and mask the area that is covered by less than $N$ support areas $S_i$. Therefore, the data pass through a layer of a neural network as if it had first passed through a masking and some information is then ignored.

*make the assumption clear*

For the convenience of theoretical analysis, we choose a simple neural network as $f$, which only contains one convolutional block and a full connection layer. The support area of each convolutional kernel in the input space is $S_i$ and the total support area is $\cup_{i \in I_S} S_i$. Here, we simplify it to a simple restriction operator, namely, each $S_i$ is a square and all $S_i$ are disjoint with each other. Our input space is the image space and our training set contains only one element zero (the empty image).

---

[1] $I_g$, $I_S$ is the index set of $\{g_i\}$ and $\{S_i\}$ respectively. They can be finite or infinite.

[2] A restriction operator is that $\forall f$, $f|_S(x) = f(x)$ if $x \in S$, else $f|_S(x) = 0$. $S$ is called the support area.

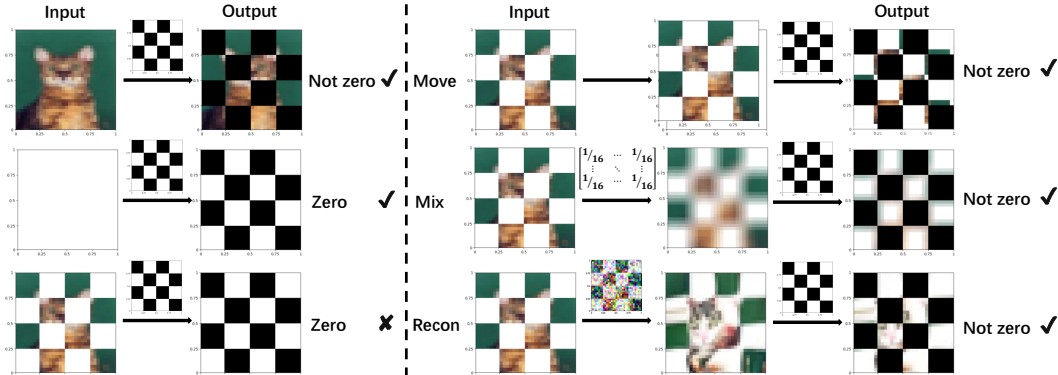

Figure 2: The left side shows the detection process of our toy example and a bad case under this setting. The right side provides three different operators to correct the result of this bad case.

Our InD is just a uniform distribution in a spherical neighborhood of zero and our target is to detect whether a new input is InD or OOD, namely, detect whether it falls in the neighborhood of zero. We show this toy example on the left side of Figure 2 and the third row shows a bad case under this setting, which looks like zero after the mask operator. On the right side of Figure 2, we use three kinds of additional operators moving, mixing and reconstruction to solve this problem.

The first two can perfectly solve the toy example in Figure 2 and can be extended to any restriction operator support on different $S$. We put the proof in Appendix A.5. The key idea of these solutions is that they move or mix information from one place to another. Then the detector $f$ can get the whole information from a small support area $S$. Such ideas can be generalized to more complex cases and the third reconstruction operator in Figure 2 plays a central role here. We find that the reconstruction strategy successfully contains the moving operator (the face of the cat moves to the left and down), the mix operator (the boundary of each small box becomes unclear) and the semantic level moving operator (the color of the cat is lighter). We can move or mix the information at both the semantic and pixel level instead of only the pixel level now. Therefore, it provides a candidate $\{g_i\}_{i \in I_g}$ with potential advantages for real detectors $f$, such as a ResNet.

### 3.3 DIFFUSION DENOISING PROCESS

Such a reconstruction strategy can be finished with any kind of generation model. However, this strategy also has its weakness. In Figure 2, we can detect the OOD examples more precisely using a reconstruction operator. However, we also need to satisfy the condition $W \subset f^{-1}(1)$, namely, to keep the pure white picture from being dirty. This challenge tells us that we need to add more control to the reconstruction strategy. In the following, we introduce the diffusion-denoising process (DDP) to solve this challenge. We dive into the structure of diffusion models and show that DDP is a kind of interpolation under the invertible condition. Therefore, DDP combines the benefit of denoising and interpolation and provides many powerful tools to control the reconstruction process. Due to space limitations, we put the analysis of enhancing invertibility in Appendix A.3 and directly analyze interesting applications of DDP under the invertible condition.

**Interpolation** Many previous papers use diffusion models to interpolate two inputs, but our method is different from the existing symmetric one using spherical linear interpolation (Shoemake, 1985). We combine the invertible diffusion models and the diffusion denoising process to get a new asymmetric interpolation. Let us assume that $x_0$ is an image and $\epsilon$ is Gaussian noise, which is the reverse of an image $x_1$, and we use $x_0$ and $\epsilon$ to get $x_t$. The diffusion-denoising process is shown in Algorithm 1 and we define DDP as $\Phi(x_t, t, 0)$. When $t$ equals zeros, we do not add any noise to the images, we can get the original image $x_0$, and when $t$ equals $T$, we remove the total image $x_0$ and only leave the noise $\epsilon$, we will do a full denoising process to this noise. Because this denoising process is invertible, we can get the original image $x_1$. Therefore, the outputs

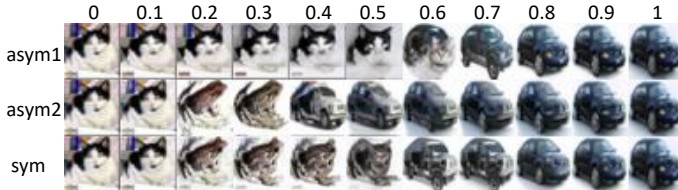

Figure 3: Different interpolation results using two fixed images.

of DDP gradually change from $x_0$ to $x_1$. Figure 3 shows the three different interpolation results. The first row uses the cat as $x_0$ and the second row uses the car as $x_0$. We can find that $x_0$ can be better preserved with respect to the symmetric interpolation in the third row.

DDP is a kind of reconstruction strategy, in the meanwhile, it is also a denoising operator and an interpolation operator. That DDP is a denoising operator means that it can keep the InD data relatively unchanged and pull the OOD data to the high-density area of InD, which is a distribution-sensitive property. That DDP is an interpolation operator means that even if we cannot reconstruct the input perfectly, we can at least control the direction of change. Both these properties give us more possibilities to control DDP and solve the above weakness. Finally, we get the following property:

*The invertible diffusion-denoising process is an asymmetric distribution-sensitive interpolation.*

### 3.4 MULTI-LAYER MODEL

In the above section, we show how to design $\{g_i\}_{i \in I_g}$ and our choice is invertible DDP. Here, we show the effect of DDP in a more complex example to design a proper detection score. We consider the last convolutional layer of a multi-layer model. In this setting, we face a major change. The input of this layer has many channels. To be more theoretical, the input feature is in $\mathbb{R}^{m \times m \times c}$, the input feature space is $\mathbb{R}^{m \times m}$ and the input feature is a combination of $c$ elements in $\mathbb{R}^{m \times m}$. Each of them is the output of a previous convolutional kernel. Therefore, a single input becomes $c$ points in the feature space instead of a single point.

make the assumption clear

For the convenience of theoretical analysis, we abstract the whole input space into 2-dimensional space and the ideal feature space belonging to the input data is the light blue square. The point in each image represents $c = 6$ features of a single input and the corresponding arrow shows the movement of these features under DDP. We use the dark blue area to show the features used by the convolutional kernels of the last convolutional layer. For better visualization, the output of each convolutional kernel is a single value, the number of points $N$ that fall in its sensitive area. Again, we assume that all the sensitive areas form a mask. This example is just a simplification of the process to compute confidence.

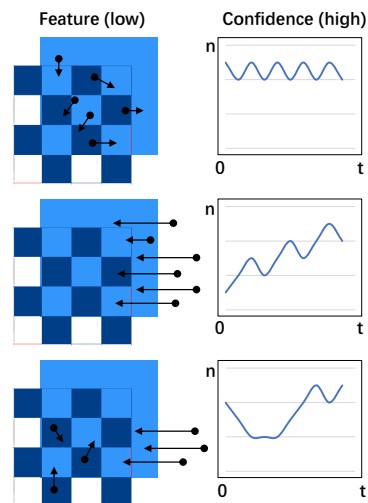

Figure 4: The change in different levels of feature space under the disturbance of DDP.

For a normal InD input (the first row), its features are uniformly distributed in the feature area, which ensures that $N$ maintains a dynamic balance under the perturbation of DDP in the feature space. For a normal OOD input (the second row), DDP pulls them to the high-density area and $N$ increases at the same time. For more challenging OOD data (the third row), it is not InD and its features are relatively sparse in the ideal feature set. However, all these features happen to fall in the dark blue area at the same time by coincidence or man-made, which causes the over-confidence problem. Such imbalance breaks the dynamic balance between the dark blue area and the remaining and causes a rapid decline of $N$ under the perturbation of DDP. Therefore, all two kinds of OOD input can be successfully detected by the change of confidence. These analyses can be extended to any middle layer by replacing the single value output with a vector output.

### 3.5 DIFFUSION-BASED NEIGHBORHOOD

The first toy example tells us that we can use DDP to enhance the detection process and the second toy example tells us what will happen in feature spaces under DDP. Detecting the dynamic change of confidence still requires the output is single-value. We can generalize this idea to detect the change in any feature space of the model by using the norm of the change as the score. After we remove the limitations of the form of the output, both DDP and the dynamic change score can be applied to any black box detector $f$, such as a ResNet.

solve the gap

The final step is to take multiple categories into account and avoid category migration for InD input. Here, we need the help of the interpolation property and there exist two choices. First, we can search

**Algorithm 1** Diffusion-denoising process

**Input:** Images $x_0$, generative interval $[0, T]$, generative gap $\delta$
1: **for** $t = T, \cdots, \delta$ **do**
2: $\quad x_t^0 = \frac{1}{\sqrt{\bar{\alpha}_t}}(x_t - \sqrt{1 - \bar{\alpha}_t}\epsilon_\theta(x_t, t))$
3: $\quad \epsilon = \epsilon_\theta(x_t, t)$
4: $\quad x_{t-\delta} = \sqrt{\bar{\alpha}_{t-\delta}}x_t^0 + \sqrt{1 - \bar{\alpha}_{t-\delta}}\epsilon$
5: **end for**
6: **Return** $x_0$

**Algorithm 2** Unconditional neighborhood

**Input:** Images $x$, timestep $t$
1: $x_{\text{knn}} = \text{KNN}(x, \{\text{training data}\})$
2: $\epsilon = \Phi(x_{\text{knn}}, 0, T)$
3: $x_{\text{noise}} = \sqrt{\bar{\alpha}_t}x + \sqrt{1 - \bar{\alpha}_t}\epsilon$
4: $x_{\text{neighbor}} = \Phi(x_{\text{noise}}, t, 0)$
5: **Return** $x_{\text{neighbor}}$

**Algorithm 3** Conditional neighborhood

**Input:** Images $x$, timestep $t$
1: $y = \text{FC}(\text{ResNet}(x))$
2: $\epsilon \sim \mathcal{N}(0, 1)$
3: $x_{\text{noise}} = \sqrt{\bar{\alpha}_t}x + \sqrt{1 - \bar{\alpha}_t}\epsilon$
4: $x_{\text{neighbor}} = \Phi(x_{\text{noise}}, y, t, 0)$
5: **Return** $x_{\text{neighbor}}$

**Algorithm 4** OOD detection

**Input:** Images $x$, diffusion-based neighborhood $x_{\text{neighbor}}$
1: $\text{fea} = \text{ResNet}(x)$
2: $\text{fea}_{\text{neighbor}} = \text{ResNet}(x_{\text{neighbor}})$
3: $\text{score} = \sum |\text{fea} - \text{fea}_{\text{neighbor}}|$
4: **if** $\text{score} > \delta$ **then**
5: $\quad$ **Return** OOD
6: **end if**

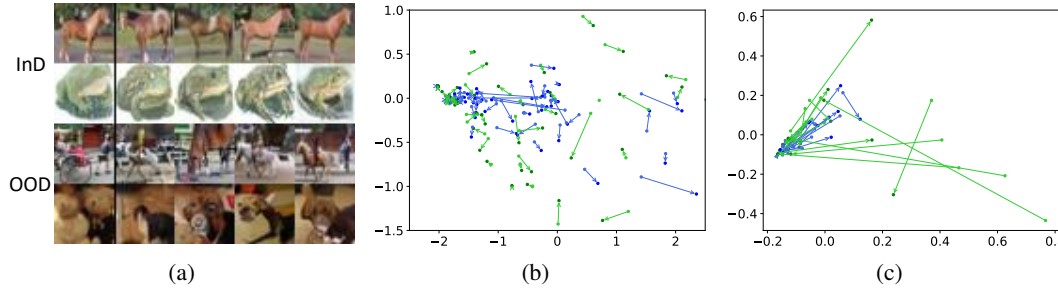

(a)            (b)            (c)

Figure 5: The influence on the different levels under DDP, including the image space, the first and the fourth feature spaces of the ResNet18. In Fig (a), the first column is the original input and the remaining is the DiffNB of them. In Fig (b, c), the blue and green arrow is the feature change of InD data and OOD data respectively under DDP after dimensionality reduction using PCA. Fig (b) and (c) show the results in the first and the fourth feature spaces of a ResNet18 respectively.

the kth nearest neighbor of the input in the input space and generate the corresponding noises of them. We interpolate these noises with the original input using DDP. Another more interesting choice is that we can train a conditional diffusion model, and fix the class condition to the class of input[3]. Then all noise are corresponding to the images in the same class. We can interpolate the input with any noise, instead of searching for it first. What's more, we can choose several noises for each input and all the results of DDP become a neighborhood of the input. We call this neighborhood the diffusion-based neighborhood (DiffNB). Then we use a discriminator model to the dynamic change in the feature spaces and determine the OOD samples based on that. We put our algorithm in Algorithm 4.

In Figure 5, we show the DiffNB of different inputs using DDP and the influence of DDP on the different feature spaces of a pre-trained ResNet18. The DiffNB has a more obvious semantic change for OOD input. For example, a rickshaw becomes a horse in Fig 5a. On the other hand, the semantic information of DiffNB of the InD data is relatively unchanged. Correspondingly, We can find that the change in low-level features (the first feature space, in Fig 5b) is similar but the change in high-level semantic features (the fourth feature space, in Fig 5c) becomes small when the input is InD and relatively big when the input is OOD.

## 4 EXPERIMENT

In this section, we first show the detailed setting of our experiments. Then we offer our OOD detection results, including our method and existing representative methods. After that, we provide ablation study results to show the contributions of each item and hyperparameter in our new scores.

---

[3]When the label is unavailable, we use the discriminator model to generate a pseudo label.

Table 1: The AUROC results of different methods. We train the models on the training data for 160k epochs and test the results on the test data. We use the conditional version method and the guidance weight is 2. We set the disturbance degree $t = 300$, the repeat size $r = 4$ and use the logic space as our detection space. The higher results are better and the bold results are the best in each case.

| InD | cifar10 | | | | | cifar100 | | | | | |
|---|---|---|---|---|---|---|---|---|---|---|---|
| OOD | cifar100 | tin | svhn | texture | place | cifar10 | tin | svhn | texture | place | avg |
| ODIN | 77.76 | 79.65 | 73.41 | 80.76 | 82.61 | 78.1 | 81.33 | 70.97 | 79.31 | 79.76 | 78.37 |
| EBO | 86.19 | 88.61 | 88.42 | 86.88 | 89.62 | 79.07 | 82.46 | 77.81 | 77.84 | 80.16 | 83.71 |
| ReAct | 86.37 | 88.91 | 89.52 | 88.19 | 90.1 | 73.48 | 79.63 | 84.45 | 83.58 | 76.94 | 84.12 |
| MLS | 86.14 | 88.53 | 88.47 | 86.89 | 89.5 | **79.18** | 82.59 | 77.68 | 77.94 | **80.29** | 83.72 |
| VIM | 87.19 | 88.86 | 97.28 | 96.03 | 90.03 | 71.54 | 78.34 | 81.15 | 87.41 | 75.77 | 85.36 |
| KNN | 89.62 | 91.48 | 95.07 | 92.84 | 91.86 | 76.48 | 83.33 | 82.09 | 83.69 | 79.03 | 86.55 |
| G-ODIN | 88.75 | 90.7 | **98.05** | 95.45 | 91.86 | 72.79 | 81.38 | 89.85 | 89.41 | 77.44 | 87.57 |
| CSI | 87.36 | 89.64 | 94.52 | 89.82 | 88.44 | 69.43 | 72.83 | 77.14 | 59.38 | 69.1 | 79.77 |
| CutMix | 85.72 | 87.99 | 90.14 | 86.51 | 90.28 | 78.6 | 82.43 | 84.05 | 77.26 | 78.53 | 84.15 |
| PixMix | **90.62** | 92.6 | 97.33 | **95.8** | 92.23 | 75.77 | 81.86 | **93.79** | **84.36** | 78.88 | **88.32** |
| Diff (ours) | 90.53 | **92.85** | 95.09 | 93.66 | **92.65** | 76.43 | **84.23** | 84.96 | 80.64 | 78.7 | 86.97 |

## 4.1 SETUP

We evaluate our methods on the most recent OOD detection benchmarks, OpenOOD benchmarks (Yang et al., 2022). We use images from six different datasets, which are filtered to ensure that the in-distribution and the OOD do not have overlapping data. We use the CIFAR10 (Krizhevsky et al., 2009) and CIFAR100 as InD samples. For the CIFAR10 dataset, we use CIFAR100, TinyImagenet (Krizhevsky et al., 2017), SVHN (Netzer et al., 2011), Texture and Places365 (Zhou et al., 2017) as OOD data. For the CIFAR100 dataset, the OOD datasets are the same, except for swapping CIFAR100 for CIFAR10 as the OOD dataset. For a fair comparison, we first train discriminator and generation models using the training set. we evaluate the results by calculating the area under the receiver operating characteristic curve (AUROC) Fawcett (2006) between the test set of the InD dataset and the test set of others, to avoid the influence of model overfitting. All images from different datasets are resized into $32 \times 32$. The discriminator models are pre-trained ResNet18 from OpenOOD. The diffusion model used in this paper is just the classical model from DDPMs. We use pre-trained unconditional models and train the conditional version by ourselves.

## 4.2 OUT-OF-DISTRIBUTION DETECTION

We put the results in Table 1. We choose ten representative baselines. The first seven methods do not adjust the discriminator model similar to our method. ODIN (Liang et al., 2018) uses temperature scaling and gradient-based input perturbation. EBO (Liu et al., 2020) uses an energy-based function. ReAct (Sun et al., 2021) uses rectified activation. MLS (Hendrycks et al., 2019) uses maximum logit scores. VIM (Wang et al., 2022) combines the information of feature space and logic space. KNN (Sun et al., 2022) uses the nearest neighbor in the feature space. All these methods are post-hoc methods and we outperform them in all cases of CIFAR10 and two cases of CIFAR100. We also compare our methods with four methods with additional training on the discriminator model. G-ODIN (Hsu et al., 2020) decomposes the posterior to model the probability of InD.

CSI (Tack et al., 2020) explores the effectiveness of contractive learning objectives. CutMix (Yun et al., 2019) and PixMix (Hendrycks et al., 2022) are two new kinds of data augmentation to improve the capability of models. Our method outperforms them in three cases and gets competitive results in the others.

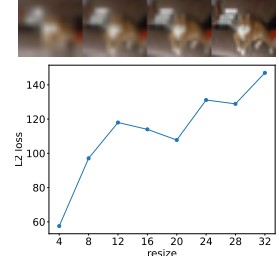

Our method performs worse than the SOTA methods when the test dataset is SVHN. A performance bottleneck is that, in addition to density estimation, DDP also has a lazy strategy in the denoising process. It tends to keep the smooth area unchanged. In Figure 6, we show this phenomenon using a simple case. We resize an InD image to $r \times r$ and then resize it back, which pulls the InD data away. However, the reconstruction error

Figure 6: The bad denoising cases for DDP.

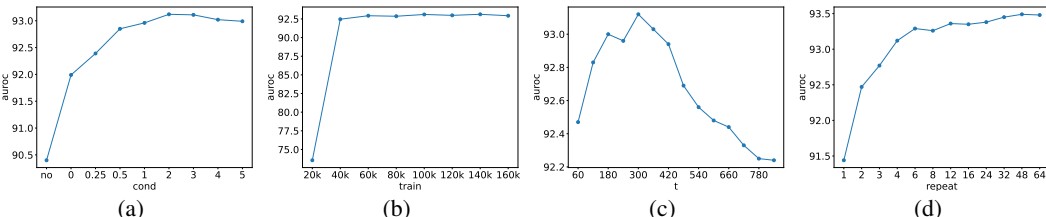

|  (a) | (b) | (c) | (d) |

Figure 8: The AUROC results under different guidance weights, training steps, disturbance degrees and resampling sizes.

decrease instead of increase under this operator. This phenomenon also occurs when the input is the relatively simple SVHN dataset.

### 4.3 ABLATION STUDY

Here, we show the influence of each item and hyperparameter on our scores. And all the setting is the same as the main experiments in Table 1 except for the ablation object.

**Detection space** In Figure 7, we compare the results when we use different detection spaces, including the input image space, the different level feature spaces, and the logit space. We get the best results when we use the high-level feature or the logit. Here, the logit has 10 dimensions and is much smaller than the high-level feature space (512 dimensions). This shows that DDP can reduce information loss successfully. The low-level features and the image space get relatively bad results, the main reason is the information is still redundant at these levels.

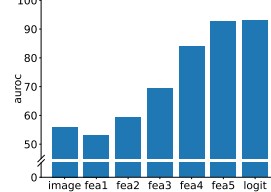

Figure 7: The results using different detection spaces.

**Condition** In Figure 8a, we compare our the unconditional and conditional methods. The main problem is that CIFAR100 has much more classes, which makes separating the feature space become more difficult and unconditional diffusion models cannot keep the interpolation in a single class. We also test different class weights $\omega$, we find that a higher class weight can get relatively better results. This shows that realness is more important than diversity in the OOD detection task. In addition, we also find that the conditional version improves the detection results on CIFAR100 more obviously, which means the conditional control is important especially when the number of classes is big.

**Training** In Figure 8b, we find that although the training process of diffusion models is relatively computation-cost to achieve the best FID results, the OOD detection does not need the models to be 100 percent well-trained (200k steps). After 40k step training, we can get relatively good results and the improvement of FID does not help the OOD detection after that.

**Timestep** In Figure 8c, we determine how to choose the best $t$ in DDP. We find that the best choice is $t = 300$ and this is consistent with our examples in Figure 4. When $t \leq 300$, the difference caused by DDP is still not obvious enough. When $t \geq 300$, the information start to lose because the noise item accounts for a larger and larger proportion, which limits the OOD detection results.

**Resampling** In Figure 8d, we determine the influence of the repeated sampling size. According to our analysis, the consistent detection results are maintained by dynamic balance, therefore, we need to resample several times to remove the random error in DDP. We find that more is better and 4 times resampling is good enough.

## 5 DISCUSSION

In this paper, we start with an assumption to explain the overconfidence problem. Then we combine discriminator and generation models to solve it. Under the setting of the first toy example, such a strategy can perfectly solve the OOD detection problem. Although we cannot say the general cases can also be perfectly solved, we show how to use this idea in the abstract feature space and get competitive results on CIFAR10 and CIFAR100 using the combination of a ResNet and a diffusion model. Our approach has good interpretability and a solid theoretical background. We believe that this strategy opens a new door to developing more powerful OOD detection methods and has the potential to be applied to OOD generalization and other related tasks.

REPRODUCIBILITY STATEMENT

Our implementation is put in the supplementary materials. We will publicize our method on GitHub once our paper is accepted.

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

# A APPENDIX

## A.1 RELATED WORK

**Diffusion Model** Denoising Diffusion Probabilistic Models (DDPMs) Ho et al. (2020) successfully generate high-quality images and make DMs become popular. For now, DMs can not only generate unconditional high-quality images but also are applied to many different fields. For the conditional generation task, DMs can do interpolation, manipulation, image-editing, style transformation and text-conditional generation Ramesh et al. (2021; 2022). For different data types, DMs can do text Austin et al. (2021), audio Kong et al. (2020); Lam et al. (2021) and video generation.

The main challenge for DMs is that they require hundreds to thousands of iterations to produce high-quality results, which limits the application of DMs. After DDPMs, many works try to make DMs faster and better. Some of them focus on the denoising equations of DMs. Nichol & Dhariwal (2021) design a better time schedule for the denoising process. Liu et al. (2022) provide new numerical methods for the denoising process. Bao et al. (2022) find analytic results for the variance of the denoising process. Some of them try to design new training strategies and new models. Salimans & Ho (2022) use distillation to accelerate DMs. Dhariwal & Nichol (2021) change the model structure from Unet to GAN to make each iteration step more powerful. What's more, Song et al. (2021b) first shows that DMs can be rewritten as two neural differential equations Chen et al. (2018); Dupont et al. (2019). Therefore, the numerical methods used to accelerate neural differential equations can also be used here.

## A.2 SIMILARITY AND DIFFERENCE

Several baselines are similar to ours in some ways. The first one is KNN (Sun et al., 2022), which does a KNN search in the feature space. However, KNN ignores the possibility that an OOD input may have a similar feature as an InD data, and all methods that only use the final feature have this problem, too. In addition, using KNN in the input space directly is also invalid, because of data sparsity and irrelevant information interference.

Another similar approach is data generation and augmentation. These methods retrain the discriminator, but our generation and discriminator models are trained separately. What's more, existing methods use generation models to generate OOD data (Marek et al., 2021) to help the discriminator models to know the capability boundary. Our methods use the generation models to do interpolation between the new input data and the training data, which do not need to carefully classify the training data and design new loss functions. Some methods use data augmentation to help the training process of discriminator models. Although classical data augmentation can enhance the richness of data and keep the data realistic, some new methods start to add complex and unreal augmentation (Hendrycks et al., 2022), which increases the burden on the models and lacks clear motivation.

Many methods combine a generation model and a discriminator model to do OOD detection, such as Ge et al. (2017); Neal et al. (2018); Lee et al. (2017); Du et al. (2022). However, we find these papers all try to use generation models to generate OOD data, which needs to design new object functions and combine the training process of the generation and discriminator models. Our method use diffusion models to do interpolation and detect the dynamic change under DDP, which is simple and effective. [more papers using generation and discriminator models]

## A.3 INVERTIBLE DIFFUSION MODEL

Here, we introduce more details of the diffusion model used in this paper. In addition, we analyze how to make the iteration of diffusion models more invertible.

**Score-based generation model** Song et al. (2021b) show that the diffusion-denoising process can also be treated as two differential equations:

$$
\begin{aligned}
dx &= (\sqrt{1 - \beta(t) - 1})x(t)dt + \sqrt{\beta(t)}dw \\
dx &= \left( (\sqrt{1 - \beta(t) - 1})x(t) - \frac{1}{2}\beta(t)s_\theta(x(t), t) \right) dt.
\end{aligned}
\tag{5}
$$

This is called probability flows (PFs). The noise $\epsilon_\theta$ of DMs and the gradient of logic likelihood $s_\theta$ are equivalent Bao et al. (2022). More specifically, we have that $s_\theta(x, t) = -\frac{1}{1-\bar{\alpha}_t}\epsilon_\theta(x, t)$.

**Pseudo numerical method** Liu et al. (2022) provide pseudo numerical methods for diffusion models (PNDMs) to accelerate DDIMs. PNDMs define Equation (3) with $\sigma_t = 0$ as transfer function:

$$\phi(x_t, \epsilon_t, t, t-\delta) = \frac{\sqrt{\bar{\alpha}_{t-\delta}}}{\sqrt{\bar{\alpha}_t}}x_t - \frac{(\bar{\alpha}_{t-\delta} - \bar{\alpha}_t)}{\sqrt{\bar{\alpha}_t}(\sqrt{(1-\bar{\alpha}_{t-\delta})\bar{\alpha}_t} + \sqrt{(1-\bar{\alpha}_t)\bar{\alpha}_{t-\delta}})}\epsilon_t. \quad (6)$$

PNDMs combine this transfer function with the noise estimated by classical numerical methods, like the linear multistep method, to get the new denoising equations:

$$\begin{cases} \epsilon'_t = \frac{1}{24}(55\epsilon_t - 59\epsilon_{t+\delta} + 37\epsilon_{t+2\delta} - 9\epsilon_{t+3\delta}) \\ x_{t-\delta} = \phi(x_t, \epsilon'_t, t, t-\delta). \end{cases} \quad (7)$$

Here, $\epsilon_t = \epsilon_\theta(x_t, t)$. Both PFs and PNDMs accelerate the denoising process without loss of quality.

**Invertibility** We show the test results in Figure 9. For DDIMs, the error occurs at the beginning, and the error accumulates with the increase of the total generation step. For PFs, the initial error is not huge, but the cumulative error occurs when the number of the total generation steps is bigger than 500. DDIMs are first-order methods, and other methods are high-order methods. We can say higher convergent order can increase the invertibility. PFs use numerical methods of adaptive step size, and PNDMs use methods of fixed step size. Therefore, we think that fixed step size can also benefit invertibility. To verify this, we replace the methods of adaptive step size used by PFs with the methods of fixed step size and call it probability flows plus (PFs+). We find that the errors decrease immediately. The reason is that fixed step

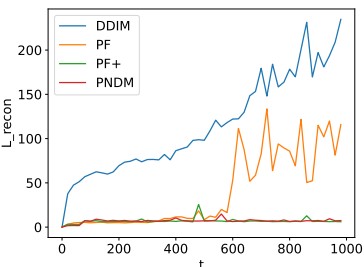

Figure 9: The reconstruction error under different iteration interval $[0, t]$ and fixed step size 20.

size maintains consistency between the sampling locations of the forward and reverse processes, which benefits the invertibility. Combining the above analysis, we have the following property:

*High convergent order and fixed iteration step size can improve the invertibility of DMs under fixed total iteration steps.*

### A.4 INTERPOLATION

In Algorithm 5 and 6, we introduce two types of interpolation using diffusion models. The positions of $x_0$ and $x_1$ are symmetric in the original interpolation and asymmetric for the asymmetric one. According to our experiment, $x_0$ is more important in asymmetric interpolation. The visualization of these interpolations can be found in Fig 3.

---

**Algorithm 5** Symmetric interpolation

**Input:** Images $x_0^1, x_0^2$, generative gap $\delta$, interpolation rate $\sigma$
1: $x_T^1, x_T^2 = \Phi(x_0^1, 0, T, \delta), \Phi(x_0^2, 0, T, \delta)$
2: $x_T^{\text{inter}} = \text{Slerp}(x_T^1, x_T^2, \sigma)$
3: $x_0^{\text{inter}} = \Phi(x_T^{\text{inter}}, T, 0, \delta)$
4: **return** $x_0^{\text{inter}}$

---

**Algorithm 6** Asymmetric interpolation

**Input:** Images $x_0^1, x_0^2$, generative gap $\delta$, interpolation timestep $t$
1: $x_T^2 = \Phi(x_0^2, 0, T, \delta)$
2: $x_t^{\text{inter}} = \sqrt{\bar{\alpha}_t}x_0^1 + \sqrt{1-\bar{\alpha}_t}x_T^2$
3: $x_0^{\text{inter}} = \Phi(x_t^{\text{inter}}, t, 0, \delta)$
4: **return** $x_0^{\text{inter}}$

---

### A.5 THE PROOF OF TOY EXAMPLE

To make this claim strict, we need to make some definitions. Each image can be represented as a function $r$ on $[0, 1]^2$ and the value of $r(x, y)$ is the RGB value at position $(x, y)$. And the images are continuous in most positions. Therefore, we simplify the input space to $\mathcal{C}([0, 1]^2)$ the continuous function on $[0, 1]^2$ and then to one-dimensional $\mathcal{C}([0, 1])$ for simplicity. The mask operator is a

restriction operator $\phi_S(r) = r|_{S=[0,0.25]\cup[0.5,0.75]}$ here. The InD is just $\{r \in \mathcal{C}([0,1]) \mid |r| \le \sigma\}$ and $|r|$ is the max absolute value of $r$ on $[0,1]$. The bad cases form a set that satisfies $\mathcal{A}_\sigma(\phi_S) = \{r|\phi_S(r) = \phi_S(0) = 0, |r| > \sigma\}$. We can prove that $\mathcal{A}_\sigma(\phi_S) = \emptyset$ when we choose proper $\{g_i\}$.

A straightforward solution is that for each $r \in \mathcal{C}(\mathbb{R})$, let $\{g(x)_i = r(x+i)|i \in \{0, \pm 0.25\}\}$, which represents the moving operator. The proof is that we can use $g_0|_S$ to get the information about $r$ on $S$ and use $g_{\pm 0.25}|_S$ to get the remaining on $[0,1]/S$. Then $r$ mush satisfies $|r(x)| < \sigma, \forall x \in [0,1]$. There also exist other kinds of solutions. For example, let $\{g(x)_{a,b} \equiv \frac{1}{b-a}\int_a^b r(x)dx|a, b \in [0,1]\}$, which represents the mixing operator. In addition, the mask operator is not necessary and we can extend it to any restriction operator.

Here we prove that $\{g(x)_i = f(x+i)|i \in \{0, \pm 0.25\}\}$ and $\{g(x)_{a,b} \equiv \frac{1}{b-a}\int_a^b f(x)dx|a, b \in [0,1]\}$ can solve the problem:

> *Given the input space $\mathcal{C}([0,1])$, the restriction operator $\phi_S$ and a fixed $\delta$, how can we design additional operator set $\{g_i\}$ to minimize the annihilator set $\mathcal{A}_\delta(\{g_i|_S\})$?*

For $\{g(x)_i = f(x+i)|i \in \{0, \pm 0.25\}\}$, $g_0|_S = f|_S$ and it equals to zero means that f is zero on S. And then we also know that $g_{0.25}|_S = f|_{S-0.25}$ and $g_{-0.25}|_S = f|_{S+0.25}$ equals zero, which means that $f$ is zero on $S - 0.25 \cup S + 0.25 = [0,1]/S$. Then the only choice of $f$ is zero and the annihilator set is empty.

For $\{g(x)_{a,b} \equiv \frac{1}{b-a}\int_a^b f(x)dx|a, b \in [0,1]\}$. If $\exists x_0 \in [0,1]$, $f(x_0) > \delta$, then $\exists \epsilon$ s.t. $\forall x \in [x_0 - \epsilon, x_0 + \epsilon]$, $f(x) > \delta$ because $f$ is continuous. Then we have that $g(x)_{x_0-\epsilon, x_0+\epsilon} \equiv \frac{1}{b-a}\int_a^b f(x)dx > \delta$ and $g(x)_{x_0-\epsilon, x_0+\epsilon}|_S > \delta$, too. This does not satisfy our condition, so the annihilator set is empty.

## A.6 OOD DETECTION RESULT

We add experiments about adversarial OOD examples in 2. We use 5000 images of CIFAR10 to generate 5000 adversarial OOD examples. We use PGD and OnePixel methods, which use $L_\infty$ and $L_0$ norms, respectively. We use an open library Harry24k/adversarial-attacks-pytorch to generate the adversarial examples. For PGD, we use eps=8/255, alpha=2/255 and steps=10. For Onepixel, we use pixels=10, steps=10, popsize=10 and inf_batch=128. We find that our method can perform pretty well on the adversarial OOD examples.

We also test the computation cost of our method. We record the total time to compute the detection scores of 5000 images using an RTX3090. We use about 0.06s to process an image. To be honest, it is much slower compared to existing methods. However, just like the success story of diffusion model acceleration, we believe that our new method can be faster in the future.

Table 2: The AUROC results of different methods. We train the models on the training data for 160k epochs and test the results on the test data. We use the conditional version method and the guidance weight is 2. We set the disturbance degree $t = 300$, the repeat size $r = 4$ and use the logic space as our detection space. The higher results are better and the bold results are the best in each case.

|         | cifar100 | tin   | pgd   | onepixel | avg      | time |
|---------|----------|-------|-------|----------|----------|------|
| ODIN    | 77.76    | 79.65 | 34.52 | 65.23    | 64.29    | 0.22 |
| EBO     | 86.19    | 88.61 | 33.13 | 71.52    | 69.8625  | 0.05 |
| ReAct   | 86.37    | 88.91 | 57.92 | 73.28    | 76.62    | 0.04 |
| MLS     | 86.14    | 88.53 | 33.56 | 71.55    | 69.945   | 0.03 |
| VIM     | 87.19    | 88.86 | 91.96 | 78.98    | 86.7475  | 0.09 |
| KNN     | 89.62    | 91.48 | 87.39 | 80.09    | 87.145   | 0.25 |
| G-ODIN  | 88.75    | 90.7  | **97.79** | 80.99 | 89.5575  | 2.31 |
| CSI     | 87.36    | 89.64 | 83.54 | 80.38    | 85.23    | 28.1 |
| CutMix  | 85.72    | 87.99 | 82.92 | 73.17    | 82.45    | 2.05 |
| PixMix  | **90.62** | 92.6  | 93.77 | 57.87    | 83.715   | 2.51 |
| Diff (ours) | 90.53 | **92.85** | 93.92 | **82.21** | **89.8775** | 340 |

