# OpenReview forum: "Out-of-distribution Detection with Diffusion-based Neighborhood"
_ICLR.cc/2023/Conference — Submitted to ICLR 2023_

### Official Review · Reviewer_sRvP · 2022-10-18

**Confidence:** 4
**Correctness:** 2
**Technical Novelty And Significance:** 3
**Empirical Novelty And Significance:** 2
**Recommendation:** 3

**Clarity, Quality, Novelty And Reproducibility:**

The paper is not written clearly such that I have to guess a lot of details of their method. The novelty is valid for me. No one uses diffusion models for OOD detection so far.

**Strength And Weaknesses:**

strength:
1. the paper uses diffusion models to replace the requirement of the real outliers in the ood detection literature. the idea makes sense to me.
2. the results seem to be effective from the main tables on one benchmark.

weakness:
1. the paper is really hard to grasp. some of the descriptions are quite vague. What does the author mean by saying "reduces the information loss of feature extraction."?
2. there are some obvious mistakes in the paper, such as the algorithmic blocks such that It is heavy to grasp the flow of the method. For example, in Algorithm 2 and 3, the x_noise is generated but it is never used in the algorithm, x_neighbor does not have a closed-form relationship with x_noise.
3. the training time is suspicious, the author trained the classification and also the diffusion model for 160k epochs, which is not scalable in practice. Also, no experiments show its effectiveness on large-scale benchmarks, such as Imagenet1k as the in-distribution data.
4. There are no visualizations provided on the generated neighboring data points so that the readers might be confused why the proposed approach works.
5. the paper ignores some existing literature on using generation for OOD detection, such as [1-4]. Some additional discussion are favorable for readers to get the context.

[1] Zongyuan Ge, Sergey Demyanov, Zetao Chen, and Rahil Garnavi. Generative openmax for multi-class open set classification. In British Machine Vision Conference 2017. British Machine Vision Association and Society for Pattern Recognition, 2017.

[2] Kimin Lee, Honglak Lee, Kibok Lee, and Jinwoo Shin. Training confidence-calibrated classifiers for detecting out-of-distribution samples. In International Conference on Learning Representations, 2018.

[3] Lawrence Neal, Matthew Olson, Xiaoli Fern, Weng-Keen Wong, and Fuxin Li. Open set learning with counterfactual images. In Vittorio Ferrari, Martial Hebert, Cristian Sminchisescu, and Yair Weiss, editors, Computer Vision – ECCV 2018

[4] Xuefeng Du, Zhaoning Wang, Mu Cai, and Yixuan Li. Vos: Learning what you don’t know by virtual outlier synthesis. In International Conference on Learning Representations, 2021

**Summary Of The Paper:**

This paper introduces diffusion models (DMs), a kind of powerful generative model, into OOD detection and finds that the denoising process of DMs also functions as a novel form of asymmetric interpolation. This property establishes a diffusion-based neighborhood for each input data. Then, the authors perform discriminator-based OOD detection based on the diffusion-based neighborhood instead of isolated data.

**Summary Of The Review:**

This paper introduces diffusion models (DMs) to generate outliers for OOD detection. The method is valid to me but further clarifications and visualizations are needed to understand the approach better. Will consider changing my score after I see some of the other reviewers' opinions and the authors' responses.

---

> ### Author Response · Authors · 2022-11-13
> **Thank you for questions and feedback**
>
> We thank the reviewer for the insightful feedback. We are delighted that our method is valid to the reviewer.  We think that the reviewer raised many useful questions, and we believe those questions have improved our work very much.
>
> Below we address specific questions and comments:
>
> 1. **[" the paper is really hard to grasp ... "]** This is a big problem and mentioned by other reviewers, too. We have reorganized the logic of Chapter 3. In the introduction, we introduce the idea behind our method using a new water detection example. In section 3.1, we theoretically define the problem and provide the main assumption. We assume that different channels concentrate on different subareas and even in the class-related area, some subareas are more important than others. In Section 3.2, we abstract this assumption to a masking operator.  The key idea is to set a threshold $N$ and mask the area that is sensitive to less than $N$ channels. Using this toy example, we show that such inequitable treatment can cause bad cases and we design additional operators to solve this problem. In Section 3.3, we develop a more powerful method to satisfy the requirements of the additional operators. In Section 3.4, we test our new method in a more complex situation and design a general detection score based on the results. In Section 3.5, we have DDP and general dynamic score and we can extend our method to any black box detector. In addition, we introduce a conditional model to solve the multi-class problem.  In Fig 5, we also visualize the actual effect of DDP when the detector is a pre-trained ResNet18. We hope these improvements will help the reviewer better understand our approach.
> 2. **["reduces the information loss ..."]** We add a new Section 3.1 to provide our theoretical setting and our main assumption. The information loss is just from our main assumption, and why generation models can solve this problem can be found in Section 3.2. The main idea is that they move or mix information from one place to another. Then the information loss can be recovered.
> 3. **["the algorithmic blocks ..." & "There are no visualizations ..."]** Sorry for the mistake. We correct the mistake in our algorithms. In addition, we add a new Section 3.5 to introduce our new method clearly. In Fig 5, we visualize the DiffNB and the change of different feature spaces under DDP.
> 4. **["the training time is suspicious ..."]** We use pre-trained ResNet18 from OpenOOD. We retrain the diffusion model similar to the setting of the original paper "Denoising Diffusion Probabilistic Models." And we find we do not need to train 160k steps (instead of epochs, sorry for the mistake) and 40k is enough for OOD detection. We agree with the reviewer that this bottleneck should be further improved.
> 5. **["the paper ignores some existing literature"]** Thanks for the reviewer's advice. We add more analysis about these papers. Due to space limitations, we put them and the existing similarity analysis in Appendix 2. We find these papers all try to use generation models to generate OOD data, which needs to design new object functions and a new training process for the generation and discriminator models. Our method use diffusion models to do interpolation and detect the dynamic change under DDP, which is relatively simple and effective.
> 6. **["Imagenet1k ..."]** It needs a lot of time to train a big diffusion model for Imagenet1k. We are working on it and will provide the results once we finish the experiment.
>
> Thanks again for the reviewer's comments on our paper. If our replies feel satisfactory, we would like to kindly ask the reviewer to consider raising the score accordingly. At the same time, we are happy to discuss any questions further.

---

### Official Review · Reviewer_ZBTr · 2022-10-23

**Confidence:** 3
**Correctness:** 3
**Technical Novelty And Significance:** 3
**Empirical Novelty And Significance:** 2
**Recommendation:** 3

**Clarity, Quality, Novelty And Reproducibility:**

This study presents a novel perspective on OOD modeling, which is quite inspiring. However, its performance is not fully validated, and the quality of this work needs to be improved while focusing on clarity, particularly with detailed explanations and revising the layout of the content significantly.

**Strength And Weaknesses:**

[+] Introducing the (noise) interpolation scheme to the OOD problem via the diffusion process has a strong mathematical background and is claimed to be more interpretable.

[+] Combining generative and discriminative models adds another proxy for distinguishing between ID and OOD. This enables the coordination of generator-sensitive and discriminator-sensitive features.

[+] The Open-OOD benchmarks allowed for a fair evaluation.

[-] The presented experimental results (in Table 1) are not very impressive compared to the previous methods.

[-] Overall, the clarity and readability of the manuscript have been compromised, and there is a crucial need for extensive revisions.

[-] Insufficient methodological clarification was given: it seems impracticable for readers to get a high-level overview of the diffusion models; rather, it is required to focus on how to organize and develop the diffusion-based neighborhood for identifying IOD and OOD. (especially how the ‘interpolation’ is managed to develop the classifier)

[-] It is not straightforward to grasp the context of Figure 2 and Figure 4. A more detailed explanation should be presented in the caption and the main text.

[-] The descriptions in Algorithms 1-3 need to be improved. For example, the input and output of each line of codes do not match with those in other lines and some notations are used without introduction.

[-] Figure 7 illustrates that the performance of the proposed strategy appeared to be hyper-parameter-sensitive. Nonetheless, Table 1 lacks a criterion for its selection.

[-] We encourage the authors to provide more implementation details and the source code to the public to guarantee reproducibility.

[Q1] How does the model function when neighbors are obtained from random noises rather than images?


**Summary Of The Paper:**

This paper proposes a new OOD detection framework that combines a typical discriminator approach with a diffusion-based neighborhood definition. Motivated by a synthetic scenario, the de-noising process of invertible diffusion models is regarded as a kind of asymmetric interpolation and is utilized to find the diffusion-based neighbors of a given input. Then the OOD samples are detected based on the features of the neighborhood. The method is validated on OOD detection benchmarks and some analyses on the model component and sensitivity on the hyperparameters are also given.

**Summary Of The Review:**

The proposed idea seems novel and could inspire the readers. However, its performance is less impressive yet, and to be officially presented at the conference, the manuscript must be strengthened in its readability, and the authors need to provide enough details to guarantee its reproducibility.

---

> ### Author Response · Authors · 2022-11-13
> **Thank you for questions and feedback**
>
> We thank the reviewer for the thoughtful feedback. We are delighted that the reviewer thinks the proposed idea is novel and inspires the readers. We think the reviewer raised many useful questions, and we believe those questions have greatly improved our work.
>
> Below we address specific questions and comments:
>
> 1. **["the clarity and readability ..."]** This is a big problem and mentioned by other reviewers, too. We have reorganized the logic of Chapter 3. In the introduction, we introduce the idea behind our method using a new water detection example. In section 3.1, we theoretically define the problem and provide the main assumption. We assume that different channels concentrate on different subareas and even in the class-related area, some subareas are more important than others. In Section 3.2, we abstract this assumption to a masking operator.  The key idea is to set a threshold $N$ and mask the area that is sensitive to less than $N$ channels. Using this toy example, we show that such inequitable treatment can cause bad cases and we design additional operators to solve this problem. In Section 3.3, we develop a more powerful method to satisfy the requirements of the additional operators. In Section 3.4, we test our new method in a more complex situation and design a general detection score based on the results. In Section 3.5, we have DDP and general dynamic score and we can extend our method to any black box detector. In addition, we introduce a conditional model to solve the multi-class problem.  In Fig 5, we also visualize the actual effect of DDP when the detector is a pre-trained ResNet18. We hope these improvements will help the reviewer better understand our approach.
> 2. **["how to organize and develop the diffusion-based neighborhood ..." & "The descriptions in Algorithms 1-3 ..."]** Thanks for the reviewer's advice. We add a new Section 3.5 to introduce our method and provide visualization of the DiffNB and the change of different feature spaces under DDP in Fig 5. We are sorry about the algorithm's mistake and have corrected them now. Thanks to the reviewer again.
> 3. **["The presented experimental results ..."]** Our method is not an improvement based on the SOTA methods. We provide a brand-new assumption and method for OOD detection. There are many spaces to improve our method further. For example, we can use some tricks to solve the problem mentioned in Section 4.2. However, it is unrelated to the main pipeline of our method, and we decide to concentrate on the effect of our main method. In addition, no baseline method can completely outperform the rest of the methods.
> 4. **["hyperparameter-sensitive ..."]** We provide three main hyperparameters for the detection process: the class weights $\omega$, the timestep $t$, and the number of sampling $N$. $\omega$ is from the classifier-free guidance. $t$ is from DDP. $N$ is from the random sampling strategy. In Section 4.3, we analyze these hyperparameters' role and find that their effect is less than 3%. The training time and the detection space are almost fixed. We want readers to have a better big picture and put them in Section 4.3, too. We put the setting of our experiment in the caption of Table 1.
> 5. **["the source code ..."]** Sorry for the misinformation. We have added a clear reproducibility statement now. We submit our code in the supplementary material when we submit our paper and mention it in the abstract.
> 6. **['How does the model function ...']** For the unconditional method, the KNN version is a little better than random noise. But both these methods are worse than the conditional method, and this method uses random noise and adds additional information by the condition input $c$.
>
> Thanks again for the reviewer's comments on our paper. If our replies feel satisfactory, we would like to kindly ask the reviewer to consider raising the score accordingly. At the same time, we are happy to discuss any questions further.

---

### Official Review · Reviewer_ziL9 · 2022-10-23

**Confidence:** 3
**Correctness:** 2
**Technical Novelty And Significance:** 2
**Empirical Novelty And Significance:** 2
**Recommendation:** 3

**Clarity, Quality, Novelty And Reproducibility:**

The paper could be better written. A lot of detail in 2.2 could be abstracted and  perhaps moved to the Appendix, since it refers to past work and doesn't add value to understanding the Toy Example or the definitions  there -- I am not sure specific equations such as (6) enhance the understanding though they may be useful in reproducing the results. Instead, the authors could use the space to provide a more detailed discussion of the definitions used in the Toy Example and intuition behind them. I have concerns over the Toy Example overall since it seems like a rare scenario. But, if you are going to provide it, it would be good to write it really well and illustrate it well with examples so that it gives a strong intuition behind your approach.

The paper started out the motivation that discriminator-based methods can be susceptible to malicious and bad cases. But, later on in the paper, in the evaluations, I did not see an analysis of how well the scheme performs on malicious examples (e.g., adversarial OOD examples that are perturbed to evade the detector). I would like to see some results on that since that was one of the motivations.









**Strength And Weaknesses:**

Strengths:
-- OOD detection is an important problem. The paper motivates their work with the point that discriminator-based methods can be susceptible to adversarial examples and bad cases.

-- The authors propose a denoising technique that diffuses an image and then does OOD classification on it.

Cons:
 -- The paper is difficult to follow due to poor writing. For instance: "Let us assume that x0 is an image and epsilon is Gaussian noise, which  is the reverse of image x1, and we ...". One has to figure out what which refers to. The noise seems wrong.
 -- Figure 4 is not clear
 --The  mathematical model could be clearer.
 -- The Toy example seems  contrived and very unlikely to occur in practice.
 -- The empirical results are not convincing. It seems that the results are mixed.
 -- It is not clear how efficient the OOD detection scheme is.  Some discussion of the efficiency of OOD detection versus other techniques may be useful.



**Summary Of The Paper:**

The paper applies a denoising function (in the form of a diffusion model) to an image before doing OOD detection. The claim is that the new method outperforms other OOD methods on two of the datasets CIFAR10 and CIFAR100.

**Summary Of The Review:**

The paper left me unconvinced of the approach or the claims. The paper could use improvements on the intuition behind the approach, presentation of the approach, and empirical results.

Update after the rebuttal:
I reviewed the changes made. I still find the writing quality to be a significant issue. For instance, I made the following comment in my original review as an example of poor writing:

*"Let us assume that x0 is an image and epsilon is Gaussian noise, which  is the reverse of image x1, and we ...". One has to figure out what which refers to. The noise seems wrong.*

I don't see that addressed. The concerns with empirical results remain as well. The adversarial examples need to be adaptive to the authors' detection pipeline and designed with those in mind. I would recommend using AutoAttack as well.

I do like the water analogy. But, the toy example still seems a bit contrived to me.

---

> ### Author Response · Authors · 2022-11-13
> **Thank you for questions and feedback**
>
> We thank the reviewer for the insightful feedback. We think that the reviewer raised many useful questions and we believe those questions have improved our work very much.
>
> Below we address specific questions and comments:
>
> 1. **["Figure 4 is not clear" & "The Toy example seems ..."]** This is a big problem and has also been mentioned by other reviewers. We have reorganized the logic of Chapter 3. In the introduction, we introduce the idea behind our method using a new water detection example. In section 3.1, we theoretically define the problem and provide the main assumption. We assume that different channels concentrate on different subareas in the input space, and even in the class-related area, some subareas are more critical than others. In Section 3.2, we abstract this assumption to a masking operator. The key idea is to set a threshold $N$ and mask the area sensitive to less than $N$ channels. Using this toy example, we show that such inequitable treatment can cause bad cases, and we design additional operators to solve this problem. In Section 3.3, we develop a more powerful method to satisfy the requirements of the additional operators. In Section 3.4, we test our new method in a more complex situation and design a general detection score based on the results. In Section 3.5, we have DDP and general dynamic score, and we can now extend our method to any black box detector. In addition, we introduce a conditional model to solve the multi-class problem. In Fig 5, we also visualize the actual effect of DDP when the detector is a pre-trained ResNet18. We hope these improvements will help the reviewer better understand our approach.
>
> 2. **["Let us assume that x0 ..."]** We add a precise algorithm in Appendix A.4 to show the pipeline of different interpolations. We also provide three different interpolation results in Fig 3 to show how our new interpolation work.
>
> 3. **["The empirical results ..."]** Our method is not an improvement based on the SOTA methods. We provide a brand-new assumption and method for OOD detection. There are many spaces to improve our method further. For example, we can use some tricks to solve the problem mentioned in Section 4.2. However, they are unrelated to the main pipeline of our method, and we decide to concentrate on the effect of our main method. In addition, no baseline method can completely outperform the rest of the methods.
>
> 4. **["A lot of detail in 2.2 ..."]** Thanks for your advice. Some paragraphs in Section 2.2 and Section 3.3 are in Appendix A.3. They are used to build invertible diffusion models, which is vital to building our asymmetric interpolation.
>
> 5. **["adversarial OOD examples ..."]** The third example in Fig 4 is similar to an adversarial one, but I think such a situation may also occur by coincidence. We are doing new experiments to generate some adversarial examples and detect them. We believe this method can be extended to adversarial defense.
> In addition, in Section 3.1, we start with the water quality detection task and try to define the OOD detection task theoretically. The reviewer can find it is similar to the definition in the adversarial defense. However, we do not assume the input is close to the training data, and then the input may belong to no class at all. Namely, OOD detection needs to consider a more wide space. We know this analysis is still limited, but we believe OOD detection has its special meaning.
>
> Thanks again for the reviewer's comments on our paper. If our replies feel satisfactory, we would like to kindly ask the reviewer to consider raising the score accordingly. At the same time, we are happy to discuss any questions further.

---

> > ### Author Response · Authors · 2022-11-15
> > **More experiment results**
> >
> > Thanks for the reviewer's advice. We add more experiment results to show the effectiveness of our method.
> >
> > **["adversarial OOD examples ..."]**  We add experiments about adversarial OOD examples. We use 5000 images of Cifar10 to generate 5000 adversarial OOD examples. We use PGD [1] and OnePixel [2] methods, which use $L_\infty$ and $L_0$ norms, respectively. We use an open library  [Harry24k/adversarial-attacks-pytorch](https://github.com/Harry24k/adversarial-attacks-pytorch) to generate the adversarial examples. For PGD, we use eps=8/255, alpha=2/255 and steps=10. For Onepixel, we use pixels = 10, steps = 10, popsize = 10 and inf_batch = 128. We find that our method can perform pretty well on the adversarial OOD examples. We are not experts in adversarial attacks, so we are happy to add additional experiments if the reviewer is interested in other adversarial sample generation schemes.
> >
> > **["It is not clear how efficient ..."]** We also test the computation cost of our method. We record the total time to compute the detection scores of 5000 images using an RTX3090. We use about 0.06s to process an image. To be honest, it is much slower compared to existing methods. However, just like the success story of diffusion model acceleration, we believe that our new method can be faster in the future.
> >
> > |             | cifar100  | tin       | pgd       | onepixel  | avg         | time |
> > | ----------- | --------- | --------- | --------- | --------- | ----------- | ---- |
> > | ODIN        | 77.76     | 79.65     | 34.52     | 65.23     | 64.29       | 0.22 |
> > | EBO         | 86.19     | 88.61     | 33.13     | 71.52     | 69.8625     | 0.05 |
> > | ReAct       | 86.37     | 88.91     | 57.92     | 73.28     | 76.62       | 0.04 |
> > | MLS         | 86.14     | 88.53     | 33.56     | 71.55     | 69.945      | 0.03 |
> > | VIM         | 87.19     | 88.86     | 91.96     | 78.98     | 86.7475     | 0.09 |
> > | KNN         | 89.62     | 91.48     | 87.39     | 80.09     | 87.145      | 0.25 |
> > | G-ODIN      | 88.75     | 90.7      | **97.79** | 80.99     | 89.5575     | 2.31 |
> > | CSI         | 87.36     | 89.64     | 83.54     | 80.38     | 85.23       | 28.1 |
> > | CutMix      | 85.72     | 87.99     | 82.92     | 73.17     | 82.45       | 2.05 |
> > | PixMix      | **90.62** | 92.6      | 93.77     | 57.87     | 83.715      | 2.51 |
> > | Diff (ours) | 90.53     | **92.85** | 93.92     | **82.21** | **89.8775** | 340  |
> >
> > [1] Madry A, Makelov A, Schmidt L, et al. Towards deep learning models resistant to adversarial attacks[J]. arXiv preprint arXiv:1706.06083, 2017.
> >
> > [2] Su J, Vargas D V, Sakurai K. One pixel attack for fooling deep neural networks[J]. IEEE Transactions on Evolutionary Computation, 2019, 23(5): 828-841.

---

### Official Review · Reviewer_H7mg · 2022-10-25

**Confidence:** 4
**Correctness:** 4
**Technical Novelty And Significance:** 1
**Empirical Novelty And Significance:** 2
**Recommendation:** 5

**Clarity, Quality, Novelty And Reproducibility:**

The usage of diffusion models for OOD is new in certain aspects, and extensive numerical experiments are presented to demonstrate good performance. 2 out of 4 datasets in OpenOOD benchmarks are conducted in the experiment.
Although this paper gives plenty of toy examples to demonstrate motivation, the readability can be improved. Some examples, like figure 4 seem very confusing.

**Strength And Weaknesses:**

Strength
- It provides a novel and promising strategy for OOD detection. Although using the generative model together with the discriminative model for OOD detection has been proposed in [1], the intrinsic idea behind how to use the generative model is radically different.
- The paper is well written, easy to follow, and with plenty of experiments to verify the effectiveness of their method.

[1] Oodgan: Generative adversarial network for out-of-domain data generation


Weaknesses:
-   The toy example in section 3.1 and section 3.3 are too disjoint with the practice, which makes the motivation not very convincing. More specifically, the discriminator considered in section 3.1 is a mask operator, which will hardly be used in practice as a discriminator. Adding some more realistic and convincing examples would be helpful.
-   Figure 4 demonstrates the dynamic of confidence over time has a different pattern for InD input and OOD input. However, their proposed algorithm does not rely on the dynamic of the confidence under the perturbance of DDP. And Figure 4 itself is also very confusing, with the differences between the three cases hard to understand. It is suggested that the author have a discussion of the performance by using the dynamic information.

**Summary Of The Paper:**

OOD detection is an important task for reliable AI. This paper proposes a new framework for OOD detection which combines the generative and discriminative models together. Specifically, they use the diffusion model for modifying the data under control and use the changes in the feature space for OOD detection. They choose ten representative methods to compare with the proposed methods on several datasets.

**Summary Of The Review:**

Overall I like the idea of applying diffusion models to do out-of-distribution detection, and this paper showed good performance compared with existing baselines. This paper can be greatly improved in terms of: more realistic and meaningful motivation examples; more detailed demonstration of the proposed method, etc.

---

> ### Author Response · Authors · 2022-11-13
> **Thank you for questions and feedback**
>
> We thank the reviewer for the thoughtful feedback. We are delighted that the reviewer likes applying diffusion models to out-of-distribution detection. We think the reviewer raised many useful questions, which have greatly improved our work.
>
> Below we address specific questions and comments:
>
> 1. **["The toy example in section 3.1 and section 3.3 ..."]** This is a big problem and has also been mentioned by other reviewers. We have reorganized the logic of Chapter 3. In the introduction, we introduce the idea behind our method using a new water detection example. In section 3.1, we theoretically define the problem and provide the main assumption. We assume that different channels concentrate on different subareas in the input space, and even in the class-related area, some subareas are more critical than others. In Section 3.2, we abstract this assumption to a masking operator. The key idea is to set a threshold $N$ and mask the area sensitive to less than $N$ channels. Using this toy example, we show that such inequitable treatment can cause bad cases, and we design additional operators to solve this problem. In Section 3.3, we develop a more powerful method to satisfy the requirements of the additional operators. In Section 3.4, we test our new method in a more complex situation and design a general detection score based on the results. In Section 3.5, we have DDP and general dynamic score, and we can now extend our method to any black box detector. In addition, we introduce a conditional model to solve the multi-class problem. In Fig 5, we also visualize the actual effect of DDP when the detector is a pre-trained ResNet18. We hope these improvements will help the reviewer better understand our approach.
> 2. **["the dynamic of confidence"]** Thanks for your advice. In Figure 4, we use a single-value output, the confidence, to show the effect of DDP. We find such dynamic change can be extended to any level feature and vector output change. Therefore, we can design detection scores using any feature spaces. More details can be found at the end of Section 3.4 and the beginning of Section 3.5. In Fig 7, we test different feature spaces of a pre-trained ResNet18. The change of the logit space is just a good representation of the dynamic change of confidence.
>
> Thanks again for the reviewer's comments on our paper. If our replies feel satisfactory, we would like to kindly ask the reviewer to consider raising the score accordingly. At the same time, we are happy to discuss any questions further.

---

### Author Response · Authors · 2022-11-13
**A general response about the core contributions**

Dear reviewers and AC,

We want to apologize for our poor writing and have almost rewritten chapter 3. We hope the current version will help the reviewer better understand our approach. In this response, we would like to mention our core contributions to the reviewers and AC.

Unlike previous papers, they directly **assume some properties about the final detection score**. For example, OOD data have high reconstruction loss and low possibility scores. **We use a much weak assumption.** We try to locate the bottleneck in the discriminator model and assume that:

```
Different channels concentrate on different subareas of the input space. Some subareas are more important than others.
```

We can transfer the problem to a restriction operator. The key idea is to **set a threshold $N$ and mask the area sensitive to less than $N$ channels.** Therefore, the data pass through a layer of a neural network as if it had first passed through a masking operator and some information is then ignored.

In Fig 2, we design additional operators to break through the information loss or the information inequality problem. The mixing and the moving operators can ideally solve the toy example, and the reconstruction operator can be generalized to more general cases. In Fig 4, we consider a more general case, the last convolutional layer of a multi-layer model. The input feature is in $\mathbb{R}^{m\times m \times c}$, the input feature space is $\mathbb{R}^{m\times m}$, and the input feature is a combination of $c$ elements in $\mathbb{R}^{m\times m}$. Therefore, a single input becomes $c$ points in the feature space instead of a single point. **Many points form a dynamic balance for InD data.** On the other hand, the OOD data face a sharp change under DDP. In Fig 4, our output is single-value, but we can extend it to any vector output by using the norm of the change in feature spaces.  In Fig 5, we also provide the influence on the different levels under DDP, including the image space, the first and the fourth feature spaces of a pre-trained ResNet18 to support our analysis.

Now, we **develop a general method DDP and a general detection score from a simple assumption**, which can be applied to any black box detector. We also develop **new asymmetric interpolation** to satisfy the condition of additional operators and **introduce classifier-free guidance** to solve the multi-class problem. To be honest, **combining the diffusion model and ResNet** may be the most unimportant contribution.

We would like to kindly ask the reviewers to consider raising the score accordingly. At the same time, we are happy to discuss any questions further.

The authors of "Out-of-distribution Detection with Diffusion-based Neighborhood"

---

### Author Response · Authors · 2022-11-16
**The deadline is approaching**

Dear reviewers and AC,

Thanks again for the great efforts and valuable comments of the reviewers and AC.

We have carefully addressed the main concerns and provided detailed responses to each reviewer. We hope the reviewers might find the responses satisfactory. As the end of the rebuttal phase is approaching, we would be grateful if we could hear the reviewers' feedback. We will be very happy to clarify any remaining points.

The authors of "Out-of-distribution Detection with Diffusion-based Neighborhood"

---

### Decision · Program_Chairs · 2023-01-20

**Decision:**

Reject

**Justification For Why Not Higher Score:**

 N/A

**Justification For Why Not Lower Score:**

 N/A

**Metareview: Summary, Strengths And Weaknesses:**

The paper introduces diffusion models into OOD detection that combines a typical discriminator approach with a diffusion-based neighborhood definition. All the reviwers find that the paper is hard to follow, and there are some obvious mistakes in the paper, such as the algorithmic blocks. The paper is below the bar of top conferences. It is better to improve the paper according to the comments of reviewers.